## **Seasonal Interplay of Water Mass Mixing and Nutrient Dynamics**

### 2 in an Arctic Fjord: A Case Study of Kongsfjorden, Svalbard

| 3  |                                                                                                                |
|----|----------------------------------------------------------------------------------------------------------------|
| 4  | Hyebin Kim <sup>1</sup> , Dukki Han <sup>2</sup> , Sang Rul Park <sup>3</sup> , Tae-Hoon Kim <sup>1*</sup>     |
| 5  |                                                                                                                |
| 6  | <sup>1</sup> Department of Oceanography, Faculty of Earth Systems and Environmental Sciences, Chonnam National |
| 7  | University, Gwangju 61186, Republic of Korea                                                                   |
| 8  | <sup>2</sup> Department of Marine Molecular Bioscience, Gangneung-Wonju National University, Gangneung 25457,  |
| 9  | Republic of Korea                                                                                              |
| 10 | <sup>3</sup> Department of Marine Life Sciences, Jeju National University, Jeju 63243, Republic of Korea       |
| 11 |                                                                                                                |
| 12 |                                                                                                                |
| 13 | *Corresponding author:                                                                                         |
| 14 | E-mail. Address: thkim80@jnu.ac.kr. (TH. Kim)                                                                  |
| 15 |                                                                                                                |
| 16 | June, 2025                                                                                                     |
| 17 | To be submitted to Ocean Science                                                                               |
| 18 |                                                                                                                |

#### **Abstract**

19

20

21

22

23

24

25

26

27

28

29

30

31

32

33

34

35

36

37

38

39

40

41

This study examined seasonal variations in water mass structure and nutrient dynamics in Kongsfjorden, a high Arctic fjord where water mass composition varies seasonally due to mixing among Atlantic Water, Polar Surface Water, and glacial meltwater. In spring, the dominance of Modified Atlantic Water (MAW) facilitated active vertical mixing, leading to relatively high, uniform nutrient concentrations throughout the water column. In summer, the enhanced influence of glacial meltwater and warmer Polar Surface Waters (PSWw) resulted in strong surface stratification and significant nutrient depletion in the upper layer. To disentangle the effects of physical mixing from biological consumption, theoretical nutrient concentrations were calculated based on a four-component water mass mixing model. The positive differences between theoretical and observed concentrations (ΔNutrient) were indicative of significant biological uptake, which accounted for substantial nutrient reductions in observed surface concentrations from spring to summer: approximately  $69 \pm 18\%$  for NOx,  $74 \pm 15\%$  for phosphate, and  $47 \pm 18\%$  for silicate. Crucially,  $\Delta$ Nutrient values served as a 'biogeochemical memory,' reflecting the cumulative net biological consumption since the spring bloom rather than just instantaneous phytoplankton biomass. These biological processes also altered nutrient stoichiometry, causing the surface nitrogen-to-phosphorus (N/P) ratio to increase from 15.0 in spring to 18.8 in summer, indicating a shift in nutrient limitation patterns. Consequently, summer surface waters transitioned toward potential co-limitation, with concentrations of phosphate ( $\sim 0.13 \pm 0.07 \, \mu M$ ) and silicate ( $\sim 1.66 \pm 0.39 \, \mu M$ ) approaching their respective limitation thresholds. These findings highlight a clear seasonal transition from a physically controlled, nutrient-replete spring to a biologically regulated, nutrient-limited summer. This understanding is crucial for predicting how Arctic fjord ecosystems, and their primary

productivity, will respond to ongoing Atlantification and increased freshwater input under climate change.

#### 1. Introduction

The Arctic marine ecosystem, which is characterized by unique and dynamic environmental conditions, is governed by the complex interaction of physical, chemical, and biological factors. Within this system, nutrient availability, which is primarily controlled by ocean currents, riverine discharge, and atmospheric deposition, plays a fundamental role in maintaining biological productivity and ecological health (Duarte et al., 2012; Galloway et al., 2004; Holmes et al., 2002; Tovar-Sánchez et al., 2010). These nutrients are particularly vital for primary production, which is the foundation of the Arctic marine food web. Ocean currents, notably Atlantic and Pacific inflows, transport essential nutrients into the Arctic Ocean, thus influencing regional primary productivity (Carmack et al., 2006; Codispoti et al., 2013; Polyakov et al., 2012; Randelhoff et al., 2020; Torres-Valdés et al., 2013). As a result, seasonal fluctuations in sea ice, solar radiation, and water column stratification drive nutrient dynamics and productivity cycles (Arrigo et al., 2014). In particular, during spring and summer, increased sunlight and meltwater often promote stratification and phytoplankton blooms (Gradinger, 1995; Hegseth & Tverberg, 2013; Leu et al., 2010; Tremblay et al., 2015).

Arctic fjords such as Kongsfjorden in Svalbard are useful areas for assessing nutrient cycling processes due to the interactions between advected ocean currents (e.g., warm, saline Atlantic Water) and local water masses (Cottier et al., 2005; Svendsen et al., 2002). The inflow of nutrient-rich Atlantic Water has been shown to play a key role in regulating nutrient supply and productivity in fjord systems, contributing to complex spatiotemporal variability (Carmack

et al., 2006; Falk-Petersen et al., 2009; Rudels et al., 2005). Understanding these dynamics, especially before and after blooms, is therefore essential for predicting how Arctic fjord ecosystems respond to environmental changes. This is because seasonal shifts in nutrient availability and plankton community structure strongly influence the region's fundamental biogeochemical processes (Sakshaug, 2004; Singh et al., 2020; Reigstad et al., 2008; Tremblay et al., 2015; Vonnahme et al., 2022). Water mass mixing significantly influences nutrient distribution in Arctic fjords (Randelhoff et al., 2020; Hodal et al., 2012; Tamelander et al., 2013; Rysgaard et al., 1999). While AW inflow can enhance productivity by supplying nutrients (Carmack et al., 2006; Torres-Valdés et al., 2013), a quantitative understanding of how physical mixing and biological processes separately contribute to seasonal nutrient depletion remains a key knowledge gap. Disentangling these effects is critical for accurately assessing the biological drivers of productivity.

The present study addresses these gaps by examining seasonal (spring/summer) variation in water mass mixing and nutrient dynamics in Kongsfjorden. Specifically, a nutrient anomaly approach (ΔNutrient) derived from a water mass mixing model to quantify the net biological impact on the nutrient inventory. Furthermore, it aims to determine the impact of these seasonal mixing patterns, notably the active vertical mixing characteristic of spring and the enhanced stratification observed in summer on nutrient concentrations. A key aspect of this study is to explore whether differences between theoretical nutrient concentrations, derived from mixing models, and actual observed nutrient levels can be effectively used to discern the influence of biological processes. Specifically, this study tests the hypothesis that the difference between theoretical (mixing-derived) and observed nutrient concentrations can effectively quantify the cumulative influence of biological processes. By comparing these observed and theoretical

nutrient levels, this study will assess the relative influence of physical mixing versus biological processes. Ultimately, this research aims to provide crucial baseline data for understanding how Arctic marine ecosystems respond to climate change, particularly in the context of warming-induced alterations to water masses and mixing dynamics within sensitive fjord environments (Wassmann, 2011).

#### 2. Materials and Methods

#### 2.1 Study Sites

Kongsfjorden, an Arctic fjord situated on the west coast of Spitsbergen, Svalbard, was used as the primary study site (Fig. 1). This fjord is approximately 20 km in length and varies in width from 4 to 10 km, reaching a maximum depth of approximately 300 m near its mouth. The hydrography in Kongsfjorden is characterized by significant freshwater input from several tidewater glaciers, a process that is more intense during the summer melt season. Furthermore, the fjord is influenced by the advection of relatively warm and saline AW transported via the West Spitsbergen Current and by the presence of colder, fresher waters of Arctic origin.

#### 2.2. Sampling and Analytical Methods

Seawater samples were collected from three discrete depths within vertical water columns using a conductivity-temperature-depth (CTD) rosette system aboard the MS Teisten (April) and the RV Helmer Hanssen (July) during 2023 in Svalbard. Sampling depths were adjusted by season to capture key hydrographic features. In spring (April), samples were collected at 0 m (surface), 20 m (mid-depth), and 50 m (deep) to represent the well-mixed water column. In

summer (July), a more stratified sampling strategy was employed to resolve the sharp vertical gradients caused by meltwater; samples were collected from 0–5 m (surface), 10–25 m (middepth, capturing the thermocline), and 50–100 m (deep). Detailed station-specific depths are provided in the caption of Figure 4. During sample collection, the salinity and temperature were measured using sensors within the CTD system. Fluorescence was measured using a CTD attached fluorometer and is presented in fluorescence-derived chlorophyll-*a* concentrations (mg/m³).

Dissolved inorganic nutrients (NO<sup>2-</sup>, NO<sup>3-</sup>, NH<sup>4+</sup>, PO<sub>4</sub><sup>3-</sup>, and Si(OH)<sub>4</sub>) were analyzed using a nutrient autoanalyzer (New QuAAtro39; SEAL Analytical, UK). For each nutrient, 50 mL of seawater was filtered through 0.7 μm GF/F filters (25 mm, Whatman Inc., Florham Park, NJ, USA). This filtration was conducted using acid-washed syringes, and the filtrate was collected in polypropylene conical tubes, which were stored at –20°C until analysis. To ensure the accuracy and precision of the nutrient analysis, certified reference materials for each nutrient were run concurrently with the samples. According to the certified reference material (KANSO Co., LTD), the analytical uncertainty was within 5% for dissolved inorganic nutrients. Hereafter, the sum of nitrate (NO<sub>3</sub><sup>-</sup>) and nitrite (NO<sub>2</sub><sup>-</sup>) is referred to as NOx, PO<sub>4</sub><sup>3-</sup> as phosphate, and Si(OH)<sub>4</sub> as silicate. This terminology is used to ensure accuracy as nitrite concentrations, while minor, were not consistently negligible. For the analysis of nutrient ratios, dissolved inorganic nitrogen (DIN) was calculated as the sum of NO<sub>2</sub><sup>-</sup>, NO<sub>3</sub><sup>-</sup>, and NH<sub>4</sub><sup>+</sup>, while dissolved inorganic phosphorus (DIP) corresponded to PO<sub>4</sub><sup>3-</sup>.

#### 2.3. Water Mass Analysis and Theoretical Nutrient Concentrations

To assess the seasonal variability in the hydrographic structure of the fjord and its influence on the distribution of nutrients, water mass analysis was conducted. The mixing ratios of the different water masses present in Kongsfjorden were calculated using observed temperature and salinity data for both spring and summer. This analytical approach was in accordance with established methodologies detailed by Miller (1950) and Tomczak (1999), which require the precise definition of characteristic end-member water types that contribute to the observed water properties within the fjord. Nutrient concentrations for the end-members were adopted from the comprehensive study of Duarte et al. (2021), which provides representative background values for the water masses advected into the Svalbard region.

In the present study, four principal end-member water types were used in the mixing model due to their characteristic presence and influence in the Arctic region and specifically in Kongsfjorden: Atlantic Water (AW), Modified AW (MAW), Polar Surface Water (PSW), and its warmer variant Polar Surface Water warm (PSWw). While glacial meltwater (GMW) is a significant source of freshwater in summer, its direct influence was simplified and incorporated into the characteristics of PSWw, which represents the warm, low-salinity surface layer. This assumption is further addressed in the discussion regarding silicate dynamics. The selection of these water types was consistent with previous hydrographic characterization of the region (Nilsen et al., 2008; Rudels et al., 2000). AW, which is defined by its relatively high temperature and salinity, originates from lower latitudes and is advected into the Arctic. MAW represents AW that has undergone significant transformation through cooling, freshening, and nutrient alteration following its entry and circulation within the Arctic system. PSW is characterized by its cold temperatures and lower salinity, typically occupying the upper layers of the water column and originating from Arctic surface processes. PSWw shares many of the

same general characteristics as PSW but is distinguished by notably warmer temperatures, often reflecting the influence of seasonal surface heating and increased meltwater input, particularly during the summer months.

The temperature–salinity (T-S) characteristics defining these end-members are detailed in Table 1 and visually represented in Fig. 2a. These definitions were carefully established based on a combination of established values from past research (e.g., Rudels et al., 2000) and an examination of the observed distribution of T-S data collected during the present study. This dual approach ensured that the defined end-members comprehensively and accurately covered the full spectrum of water types observed in Kongsfjorden during the sampling periods. Because the hydrographic properties of the deep-water masses in Kongsfjorden exhibited minimal temporal variation between the spring and summer seasons, a single, consistent set of T-S characteristics for each end-member was employed for water mass analysis in both the spring and summer datasets, allowing for a direct comparison of seasonal shifts in their relative contributions.

The fractional contribution of AW, MAW, PSW, and PSWw (denoted as A, B, C, and D, respectively, Table. 1) to any given water sample (P) collected within the fjord was calculated using a standard four end-member mixing model (Fig. 2b). This model operates on the principle of the conservative mixing of temperature and salinity (Miller, 1950). The output of this model provides the fractional contributions ( $f_A$ ,  $f_B$ ,  $f_C$ , and  $f_D$ ) of each end-member to the sampled water under the fundamental constraint that the sum of these individual fractions equals unity (i.e.,  $f_A + f_B + f_C + f_D = 1$ , or 100%).

Theoretical nutrient concentrations ( $Nutrient^*$ ) for each sample were calculated by multiplying the fraction of each end-member water mass (defined in Table 1) by its end-member nutrient concentration ( $Nut_x$ ) and summing the contributions as follows:

Nutient\* = 
$$(f_A \times Nut_A) + (f_B \times Nut_B) + (f_C \times Nut_C) + (f_D \times Nut_D)$$

To assess the biological impact on nutrient concentrations, the difference (ΔNutrient)
between the theoretical and observed concentrations was calculated:

178 
$$\Delta Nutrient = Nutrient^* - Nutrient_{observed}$$

A positive value indicated net nutrient removal beyond physical mixing, which was attributed to the net biological effect, primarily biological consumption.

#### 2.4. Uncertainty Assessment

To evaluate the robustness of these calculations, a sensitivity analysis was performed to quantify the uncertainty propagated from the end-member nutrient definitions. The end-member concentrations for NOx, phosphate, and silicate were varied by  $\pm 10\%$ . This range was selected as a conservative estimate of natural variability, supported by regional and global oceanographic studies that report nutrient concentrations in major water masses to generally vary within 5–15% of the mean (Torres-Valdés et al., 2013; Hopwood et al., 2020). The resulting range in the calculated  $\Delta$ Nutrient values was used to define the uncertainty of the model-derived results, which is reported alongside the key quantitative findings. This

assessment provides a measure of confidence in the conclusions against potential variations in the end-member characteristics.

#### 2.5. Statistical analysis

All statistical analyses were performed using SPSS ver. 19 (IBM Corp., Armonk, NY, USA). Prior to hypothesis testing, the normality of the data was assessed using the Shapiro–Wilk test. Depending on the results of the normality test, either independent samples t-tests (for normally distributed variables) or Mann–Whitney U tests (for non-normally distributed variables) were applied to compare differences between groups. A significance level of p < 0.05 was used for all tests.

#### 3. Results and Discussion

#### 3.1. Seasonal Variation in Hydrography and Observed Dissolved Inorganic Nutrient

#### Levels

Kongsfjorden exhibited distinct seasonal hydrographic conditions during the study period (Fig. 3). Water temperatures in the fjord ranged from a minimum of -0.86°C to a maximum of 6.88°C (Fig. 3a), and salinity ranged from a minimum of 28.05 to a maximum of 34.93 (Fig. 3b). The spring season was characterized by lower temperatures, with a mean temperature of  $0.16 \pm 0.56$ °C, and relatively high and uniform salinity, averaging  $35.67 \pm 0.28$ . In contrast, summer had significantly warmer waters (mean:  $3.56 \pm 1.49$ °C) and markedly lower and more variable salinity (mean:  $33.03 \pm 1.92$ ). These hydrographic changes were primarily driven by

seasonal increases in solar radiation, sea ice meltwater, and glacial freshwater input, which collectively enhanced the vertical stratification of the water column.

Consistent with these hydrographic shifts, the levels of dissolved inorganic nutrients also exhibited strong seasonal patterns. The NOx concentration varied from 0.67  $\mu$ M to 10.41  $\mu$ M (Fig. 3c). During spring, the mean surface nitrate level was 7.10  $\pm$  1.83  $\mu$ M. In summer, however, mean surface nitrate concentrations decreased significantly to 2.20  $\pm$  1.15  $\mu$ M, representing an approximate 69  $\pm$  18% reduction from spring levels. While surface nitrate was depleted, concentrations in deeper water remained high, resulting in a stronger vertical gradient in summer compared to that in spring. This suggests that active vertical mixing replenished surface nutrients in spring, whereas reduced mixing and significant biological uptake occurred during the summer period.

Phosphate concentrations ranged from 0.07  $\mu$ M to 0.70  $\mu$ M (Fig. 3d). The spring surface mean was 0.50  $\pm$  0.12  $\mu$ M, declining considerably to 0.13  $\pm$  0.07  $\mu$ M during summer, a reduction of approximately 74  $\pm$  15%. Notably, summer phosphate concentrations often fell below the 0.20  $\mu$ M threshold commonly regarded as limiting for phytoplankton growth in Arctic waters (Tremblay et al., 2015). Thus, there was a strong likelihood of phosphate limitation during this period, particularly given that phosphate declined at a greater rate than nitrate from spring to summer.

Silicate concentrations ranged from 0.83 to 4.45  $\mu$ M (Fig. 3e). The mean surface concentration was 3.11  $\pm$  0.72  $\mu$ M in spring, decreasing to 1.66  $\pm$  0.39  $\mu$ M in summer, representing a 47  $\pm$  18% reduction. The summer surface silicate concentration approached the 2  $\mu$ M threshold frequently cited as indicative of potential silicate limitation for diatom growth

(Egge & Aksnes, 1992). In some samples, the summer surface silicate concentrations were higher than expected despite biological uptake, likely due to the influence of glacial meltwater enriched in silicate via bedrock erosion (Hawkings et al., 2017).

Statistical analysis confirmed that the seasonal differences observed for all three nutrients were significant (p < 0.05 for all comparisons). These observed nutrient patterns in Kongsfjorden were largely consistent with findings from previous studies in the same location (e.g., Leu et al., 2015). However, the background nutrient levels observed in this study were generally higher than those reported for some other Arctic regions, such as Young Sound, Greenland (Rysgaard et al., 1999), a difference attributable to the stronger and more direct influence of nutrient-rich AW in the Svalbard region. Spatial differences were also apparent within Kongsfjorden; in particular, stations with higher contributions from PSWw exhibited more pronounced summer surface nutrient depletion, particularly for phosphate, which had mean concentrations as low as  $0.08 \pm 0.03 \,\mu\text{M}$ . (This observation will be further discussed in the context of nutrient limitation in Section 3.4). This likely reflects the influence of glacial meltwater input and enhanced stratification associated with PSWw-dominated surface layers.

#### 3.2. Seasonal Characteristics of Water Masses and Theoretical Nutrient Concentrations

The four-component end-member mixing model revealed distinct seasonal distributions of water masses within Kongsfjorden (Fig. 4). Overall, MAW, with a mean contribution of  $52 \pm 29\%$ , and AW ( $20 \pm 16\%$ ) were the dominant water masses influencing the fjord. These water masses are recognized as the primary sources of inorganic nutrients in this system. The contributions of PSW ( $14 \pm 13\%$ ) and PSWw ( $14 \pm 14\%$ ) were lower on average, though their influence varied considerably with season and depth.

During the spring season, the proportion of MAW was generally higher throughout the water column than in summer, suggesting the active mixing of the inflowing AW and the resident PSW. This mixing is facilitated by physical and chemical processes in the Arctic Ocean that promote the formation of MAW (Rudels et al., 2004), resulting in a relatively uniform vertical distribution of water masses from the surface to the deep layers of the fjord. In contrast, the summer season was characterized by a marked shift in the water mass composition. The surface layer (0-30 m) had a considerably higher proportion of PSWw (33  $\pm$  25%) and PSW (19  $\pm$  16%), which was primarily associated with seasonal sea ice meltwater, surface warming, and freshwater-induced stratification. However, the deep layer (>50 m) remained dominated by AW (36  $\pm$  3%) and MAW (53  $\pm$  3%). This vertical stratification limited the vertical exchange of water and nutrients between the surface and deep layers.

The observed water mass distribution patterns were broadly consistent with previous descriptions of Kongsfjorden by Svendsen et al. (2002) and Cottier et al. (2005). However, the proportion of MAW observed in this study was substantially higher than reported in some earlier studies, which may reflect the ongoing process of Atlantification, which is the enhanced penetration of Atlantic-origin waters into the Arctic Ocean (Polyakov et al., 2017), or be the result of long-term changes in the Arctic hydrography and climate. This trend suggests that future warming could further intensify the influence of warm, saline Atlantic-origin waters, fundamentally altering the fjord's stratification and nutrient supply regimes. Additionally, continued glacier melting driven by regional warming is expected to increase the volume of PSWw, thus intensifying surface stratification in the future.

## 3.3. Biological Impact on Nutrient Concentrations: Differences between Observed and

**Theoretical Concentrations (ΔNutrient)** 

To assess the biological influence on nutrient dynamics, the observed nutrient concentrations were compared to theoretical values derived from end-member mixing (Fig. 5). The difference represents net nutrient removal that exceeds a level that can be explained by physical mixing alone. A positive  $\Delta$ Nutrient value indicates that observed concentrations are lower than expected from conservative mixing, thus suggesting biological uptake or transformation. With few exceptions, observed nutrient concentrations were significantly lower than theoretical values (p 

**Figure 2.** (a) Temperature–salinity (T–S) diagram showing the four end-member water types used in this study: Atlantic Water (AW), Modified Atlantic Water (MAW), Polar Surface Water (PSW), and warm Polar Surface Water (PSWw). These end-members were defined based on previously published criteria (e.g., Rudels et al., 2000) and supported by hydrographic data collected during the cruises (Table 1). (b) Conceptual diagram of the four-end-member mixing framework. Point P denotes an arbitrary water parcel in T–S space. Its location relative to the end-members was used to estimate fractional contributions ( $f_A$ ,  $f_B$ ,  $f_C$ , and  $f_D$ ), with the sum constrained to unity ( $f_A + f_B + f_C + f_D = 1$ ).

**Figure 3.** Vertical profiles of (a) temperature (°C), (b) salinity, and (c) nitrate ( $\mu$ M), (d) phosphate ( $\mu$ M), and (e) silicate levels ( $\mu$ M) in Kongsfjorden. Black circles indicate spring data; red circles indicate summer data. Data represent measurements from multiple stations and depths.

**Figure 4.** Relative contributions of the four end-member water masses (AW, MAW, PSW, and PSWw) in Kongsfjorden during (a) spring and (b) summer based on the four-component mixing model. Labels on the x-axis indicate the sampling station followed by the relative sampling depth: S (Surface), M (Mid-depth), and D (Deep). For the spring cruise (a), S, M, and D samples were typically collected at 0 m, 20 m, and 50 m, respectively (except for station A2, where D was 100 m). For the summer cruise (b), sampling depths varied by station, with S samples from 0-5 m, M from 10-25 m, and D from 50-100 m for all stations except J1. At station J1, S, M, and D samples were collected at 2 m, 5 m, and 20 m, respectively.

Figure 5. Differences between theoretical (mixing-derived) and observed nutrient concentrations ( $\Delta$ Nutrient = Theoretical – Observed;  $\mu$ M) during (a) spring and (b) summer. Bars represent  $\Delta$ Nutrient values for nitrate, phosphate, and silicate, as indicated in the legend.

**Figure 6.** Vertical profiles of (a) Chlorophyll-a (mg/m³), (b)  $\Delta$ NOx ( $\mu$ M), and (c)  $\Delta$ Phosphate ( $\mu$ M), (d)  $\Delta$ Silicate ( $\mu$ M). Black circles indicate spring and red circles indicate summer. The legend applies to all panels.  $\Delta$ Nutrient values are calculated from differences between observed and theoretical concentrations derived from end-member mixing.

**Figure 7.** Relationships between chlorophyll-*a* (mg/m³) and ΔNutrient (μM) in Kongsfjorden: (a–c) spring, (d–f) summer. Regression lines and r² values are shown for each panel.

# **Figure 8.** Relationships between salinity and $\Delta$ Nutrient ( $\mu$ M) in Kongsfjorden: (a–c) spring, (d–f) summer. Regression lines and $r^2$ values are shown for each panel.

**Figure 9.** Vertical profiles of the DIN/DIP ratio in Kongsfjorden. Black circles represent spring values; red circles represent summer. The vertical dashed line indicates the Redfield ratio (16:1).

**Table 1.** Temperature (°C), salinity, and nutrient concentrations (NOx, phosphate, and silicate; μM) for the four end-member water types: Atlantic Water (AW), Modified Atlantic Water (MAW), Polar Surface Water (PSW), and warm Polar Surface Water (PSWw). The temperature-salinity (T-S) definitions were adopted from Rudels et al. (2000), with σ<sub>0</sub> represents the potential density anomaly referenced to 0 dbar. determined based on the characteristics of the most representative samples collected in this study (identified at the vertices of the T-S diagram in Fig. 2a). Nutrient values for each water mass are based on literature values from Duarte et al. (2021).

| End<br>member | Water<br>mass                               | NOx<br>(µM) | Phosphate (µM) | Silicate<br>(µM) | Temperature (°C) | Salinity | Reference<br>(Rudels et al. 2000)                                                                                                                                          |
|---------------|---------------------------------------------|-------------|----------------|------------------|------------------|----------|----------------------------------------------------------------------------------------------------------------------------------------------------------------------------|
| A             | Atlantic<br>Water<br>(AW)                   | 10.66       | 0.82           | 4.86             | 8.2              | 35.6     | $\begin{array}{l} 27.70 < \sigma_0 < 27.97, \\ T > 2^{\circ}\mathrm{C, or} \ 27.97 < \sigma_0, \\ and \ \sigma_{0.5} < 30.444, \ T > 0^{\circ}\mathrm{C} \end{array}$      |
| В             | Modified<br>Atlantic<br>Water<br>(MAW)      | 10.55       | 0.78           | 4.94             | -0.86            | 34.95    | $27.70 < \sigma_0 < 27.97, T < 0^{\circ}\text{C},$ $S < 34.676 + 0.232 \cdot$ $T, \text{ or } 27.97 < \sigma_0, \text{ and } \sigma_{0.5} < 30.444, T > 0^{\circ}\text{C}$ |
| С             | Polar<br>Surface<br>water<br>(PSW)          | 6.91        | 0.56           | 3.85             | -1.1             | 32.8     | $27.70 > \sigma_0$ , $T < 0$ °C                                                                                                                                            |
| D             | Polar<br>Surface<br>water<br>warm<br>(PSWw) | 4.83        | 0.38           | 2.33             | 5.94             | 28.05    | $27.70 > \sigma_0$ , $T > 0$ °C                                                                                                                                            |