# Peer review of "Seasonal Interplay of Water Mass Mixing and Nutrient Dynamics"

_EGUsphere, 2025_

## Author Comment (AC2)

**Figure S1.** Relationship between salinity and (a) observed silicate concentration and (b) ΔSilicate in surface waters of Kongsfjorden during summer 2023.

[Figure]

---

## Author Response (AR1)

**#Referee 1**

**General overview:**

This was an interesting article about shifts in nutrient limitations in a fjord. It was focused on water mass characterization and mixing, showing potential impacts of biological processes on nutrient ratios. The conclusions suggested the increasing influences of Atlantic water in the region, especially in the context of future warming.

**Our Response:**

Dear Referee #1 of Ocean Science,

Thank you for your valuable feedback and the opportunity to revise our manuscript, "Seasonal Interplay of Water Mass Mixing and Nutrient Dynamics in an Arctic Fjord: A Case Study of Kongsfjorden, Svalbard" (OS-2025-2845). We are grateful to the Referee #1 for their insightful and constructive comments, which have helped us to improve the quality and clarity of our paper.

We have carefully addressed all the points raised by the Referee #1. Below is a point-by-point response to the comments detailing the changes made in the revised manuscript.

Thank you for your encouraging words and positive assessment of our work. We are pleased that you found our study to be a valuable contribution.

**Specific comments:**

1. the article suggested a shift to phosphate limitation, and in this case the phosphate in Fig. 5 could be shown on a secondary axis as this would highlight changes in P concentrations, which are obviously lower than N or Si.

**Our Response:**

This is an excellent suggestion for improving the data visualization. We agree that the changes in  $\Delta Phosphate$  were not clearly visible due to the scale difference with  $\Delta NOx$  and  $\Delta Silicate$ . As

suggested, we have revised Figure 5 to include a secondary y-axis on the right for

Δ**Phosphate.** This revision now effectively highlights the seasonal and spatial variations in phosphate anomaly, which strongly supports our discussion on the shift toward phosphorus limitation (Lines 638-640).

2. The authors do admit that glacial silicate sources are excluded, but it would be interesting to know more about this.

**Our Response:**

Thank you for this insightful comment. We agree that a more detailed discussion on the treatment of glacial silicate input is crucial for interpreting our results. We have revised the manuscript to provide a clearer rationale, supported by relevant literature.

Our decision not to include glacial meltwater (GMW) as a discrete fifth end-member was based on two primary challenges:

- 1. High and Unpredictable Variability of the Source: Defining a stable and representative silicate concentration for a GMW end-member is scientifically challenging. Studies focusing on Svalbard's tidewater glaciers, including those that discharge into Kongsfjorden, report a wide and highly variable range of silicate concentrations in summer runoff. For instance, values typically range from 2 μM to 6 μM (e.g., Nowak & Hodson, 2014; Hatton et al., 2019; Hopwood et al., 2016). This variability is driven by complex factors like subglacial water residence time, watershed lithology, and melt dynamics. Choosing any single value within this wide range would introduce a significant, unquantifiable bias into our mixing model.
- Model Parsimony and Robustness: Adding a fifth, highly variable end-member would not
  only increase the model's complexity but also its associated uncertainty, potentially reducing the
  robustness of the calculated contributions from the other, better-constrained water masses (AW,
  MAW, PSW).

Therefore, we adopted a more conservative and scientifically defensible approach by subsuming the freshwater influence into our Polar Surface Water warm (PSWw) end-member. We explicitly acknowledge that this methodological choice means our calculated  $\Delta$ **Silicate values** inherently underestimate the true biological consumption.

Crucially, this limitation strengthens our overall conclusion. Our own data show a strong inverse correlation between observed silicate and salinity in summer ( $r^2 = 0.94$ ), empirically confirming a significant, non-conservative freshwater source of silicate. The fact that we still calculate a substantial biological silicate drawdown even with a model that systematically underestimates it provides powerful and compelling evidence that biological uptake is the dominant process regulating silicate dynamics in Kongsfjorden during the summer, far outweighing the effects of physical mixing alone.

We have revised **Section 3.3** (now lines 326-338 on page 16) to include this detailed rationale and a discussion of the literature. We believe this clarification significantly strengthens the manuscript and addresses your concern directly.

3. It would also be helpful to see the full T/S from the CTD sensor data, to make the water mass definitions clearer (in Fig. 2a).

**Our Response:**

Thank you for your thoughtful suggestion regarding the T-S diagram. We appreciate your perspective on providing a broader hydrographic context to strengthen the manuscript.

We would like to clarify the specific scope and objective of our study. Our primary goal is not to provide a comprehensive physical oceanography of Kongsfjorden, but rather to quantify the relative impacts of physical mixing versus biological processes on the **nutrient dynamics within the discrete** water samples we collected.

To this end, the T-S diagram in Figure 2a intentionally and exclusively displays the data points that are **directly paired with our nutrient measurements**. We firmly believe that this approach is the

most transparent and scientifically rigorous way to present the foundation for our biogeochemical model, as it shows the exact hydrographic properties of the water for which all subsequent calculations were made.

While adding the full continuous CTD data would certainly illustrate the broader physical setting, we are concerned that it would shift the focus away from the central hypothesis of our paper. Including a vast amount of data not directly tied to our nutrient analysis could dilute the clarity of our core argument, which is about disentangling processes at our specific sampling locations.

Therefore, we maintain that the current version of Figure 2a is the most appropriate representation for the stated goals of this study. It accurately reflects the precise data set used for our model and analysis. To avoid any potential misunderstanding for the reader, we will, however, revise the figure caption to state more explicitly that the points represent the complete set of discrete samples analyzed for nutrients in this study. We hope this explanation clarifies our rationale for maintaining the current figure, and we trust that our approach is understood within the specific context of our research questions.

**Technical comments:**

1. In addition to the comments relating to Fig. 2 and Fig 5. There are some minor technical points to consider such as consistency within the references - in most cases the date is the last part of the reference, with no brackets (e.g.: Wassmann, P., Title, Ref, 2011). However, on some occasions the (more usual) method of 'author (date)' formatting has been used e.g.: Miller, Arthur (1950) though this particular example is the only time that the forename is written in full.

**Our Response:**

Thank you for pointing out these inconsistencies. We have thoroughly reviewed and reformatted the entire reference list to ensure a consistent style, following the guidelines for the journal *Ocean Science*. The specific issues you noted, such as the formatting of the Miller (1950) reference, have been corrected (Lines 551-552).

2. Another inconsistency is in the use of DIN/DIP in Fig.8, which does not match the caption. On the whole this is a very readable article and a valuable contribution to literature.

**Our Response:**

Thank you for catching this oversight. To ensure clarity and consistency with standard oceanographic terminology, we have revised the manuscript to use "DIN/DIP ratio" uniformly throughout the text (Section 3.4), in the caption for Figure 8, and on the figure's axis label. This change ensures that the terminology is consistent across all parts of the manuscript.

We believe that these revisions have addressed all your concerns and have substantially strengthened the manuscript. We thank you once again for your constructive feedback (Lines 653-655).

**#Referee 2**

This manuscript presents a carefully executed and scientifically robust investigation into seasonal changes in water mass mixing and nutrient dynamics in Kongsfjorden, Svalbard. The authors successfully apply a four-end-member mixing model to disentangle the relative roles of physical mixing and biological processes in shaping nutrient distributions, a methodological approach that is both appropriate and insightful. A key strength of the study is the introduction and effective use of the  $\Delta$ Nutrient metric, which serves as a novel and compelling proxy for cumulative biological uptake—a concept aptly framed as "biogeochemical memory." This framework not only captures seasonal transitions in nutrient regimes but also offers broader utility for interpreting biological dynamics in polar marine systems. Overall, the study is timely, highly relevant, and makes a valuable contribution to our understanding of Arctic fjord biogeochemistry in the context of accelerating Atlantification and glacial meltwater inputs under climate change.

However, there are several issues that warrant further consideration before publication.

**Our Response:**

Dear Referee #2 of Ocean Science,

Thank you for your valuable feedback and the opportunity to revise our manuscript, "Seasonal Interplay of Water Mass Mixing and Nutrient Dynamics in an Arctic Fjord: A Case Study of Kongsfjorden, Svalbard" (OS-2025-2845). We are grateful to the Referee #2 for their insightful and constructive comments, which have helped us to improve the quality and clarity of our paper.

We have carefully addressed all the points raised by the Referee #2. Below is a point-by-point response to the comments, detailing the changes made in the revised manuscript.

**Main comments**

1. The authors assume that glacial meltwater (GMW) is adequately represented within the PSWw end-member. This simplification is questionable, particularly given that silicate shows clear glacial signatures (e.g., strong salinity-silicate correlation). It might be worthwhile to treat GMW as a separate fifth end-member in the mixing model or at least conduct sensitivity analyses showing the effect of its exclusion.

**Our Response:**

We sincerely thank the Referee #2 for this critical and insightful comment, which prompted us to reexamine our data and ultimately strengthen our analysis. We fully agree that the influence of Glacial Meltwater (GMW) is a crucial aspect of Kongsfjorden's biogeochemistry.

Upon re-evaluation, we discovered an error in our originally plotted figure showing the relationship between salinity and  $\Delta$ Silicate. The corrected analysis reveals that salinity and  $\Delta$ Silicate actually exhibit a positive correlation in summer. This positive trend suggests that higher salinity waters (i.e., those less influenced by freshwater) are associated with greater biological silicate consumption relative to their initial mixed concentrations. This finding strongly supports our central hypothesis that biological uptake is a dominant process throughout the fjord.

At the same time, we confirm that the relationship between salinity and the observed silicate concentration itself (not  $\Delta$ Silicate) shows a strong negative correlation (Figure S1, r2=0.53). This observation unequivocally demonstrates that GMW is a significant source of silicate to the fjord.

This presents a clear picture: while GMW continuously supplies silicate, there is a concurrent and powerful biological drawdown occurring across all water masses. This brings us back to the Referee #2's core point about including GMW as a fifth end-member. Our decision not to do so, despite the clear evidence of its influence, is based on two primary challenges:

- 1) **High and Unquantifiable Variability:** The primary reason is the high uncertainty associated with defining a stable silicate concentration for GMW. As extensively documented (e.g., Hopwood et al., 2020; Hawkings et al., 2017), GMW silicate levels vary dramatically depending on subglacial hydrology, water-rock interaction times, and the specific glacial system. Using a single, fixed value for such a variable source would introduce a significant, and likely larger, source of error into our model.
- 2) Model Parsimony and Robustness: Adding a fifth, highly uncertain end-member would increase the model's complexity and potentially reduce the robustness of the calculated contributions from the other, better-constrained water masses (AW, MAW, PSW, and PSWw).

Therefore, we have chosen a more conservative and defensible approach. We will revise the manuscript (Section 3.3) to explicitly discuss this methodological choice. We acknowledge that by not modeling GMW as a separate end-member, our calculated  $\Delta$ Silicate values likely represent a conservative, lower-bound estimate of the true biological consumption. We argue that the fact we still calculate a substantial biological drawdown, even with a model that systematically underestimates the total silicate input from GMW, strengthens, rather than weakens, our conclusion about the importance of diatom-driven uptake. This revised discussion provides a more transparent interpretation of our findings, fully acknowledging both the influence of GMW and the limitations of our model (Lines 352-392).

2. The study spans two time points (spring and summer) in a single year (2023), which limits its generalizability regarding interannual variability. The authors should explicitly acknowledge this as a limitation and suggest future work with multi-year seasonal coverage.

**Our Response:**

We agree that this is an important limitation of our study. To address this, we will revise the Conclusion (Section 4) to explicitly state that our dataset represents a detailed "snapshot" of a single year (2023). We will also emphasize that interannual variability may substantially influence the

observed patterns and suggest that future studies incorporate multi-year, high-frequency monitoring to better capture long-term trends in this dynamic environment (Lines 454-460).

3. The calculation of theoretical nutrient concentrations assumes conservative behavior of water mass properties, but nutrients are often influenced by remineralization and benthic fluxes, especially in fjords. The authors should discuss the validity of this assumption more critically in the methodology or discussion. I would also encourage them to take a look at the following article in which the authors expanded the OMP analysis to include processes like photosynthesis - thus directly assessing the biological influence on their analysis (which is also in a similar environment):

Dinauer, A., & Mucci, A. (2018). Distinguishing between physical and biological controls on the spatial variability of pCO2: A novel approach using OMP water mass analysis (St. Lawrence, Canada). *Marine Chemistry*, 204, 107-120.

**Our Response:**

This is an excellent point. We thank the Referee #2 for encouraging a more critical assessment of our model's assumptions. We will revise the Discussion (Section 3.3) to more thoroughly address the roles of non-conservative processes like remineralization and benthic fluxes.

1. Validity of ΔNutrient in the Euphotic Zone: We now explicitly clarify that our ΔNutrient metric represents the net outcome of all non-conservative processes occurring within a water parcel. While this includes both biological uptake and remineralization, we argue that ΔNutrient remains a robust proxy for net community production in the euphotic zone for two key reasons. First, during the highly productive summer, nutrient uptake rates by phytoplankton are expected to far exceed *in-situ* remineralization rates. Second, the strong surface stratification evident in our data (Fig. 3) acts as a physical barrier, effectively isolating the surface layer from deeper, remineralized nutrient pools and from the influence of benthic fluxes. We have added this detailed justification to the manuscript to clearly define the context and validity of our approach.

We also add the caveat that  $\Delta$ Nutrient values in deeper waters could be influenced by remineralization, potentially underestimating the total downward export of organic matter.

2. Applicability of Advanced Methods (Dinauer & Mucci, 2018): We sincerely thank the Referee #2 for the reference to Dinauer & Mucci (2018). Their extension of Optimum Multiparameter (OMP) analysis to include biogeochemical processes is a powerful and elegant approach. We carefully considered its applicability to our study. However, implementing such a process-based OMP model is contingent upon having a sufficient number of independent, conservative tracers to constrain the system. To resolve *n* sources and processes (in our case, 4 water masses + biological processes), the model requires at least *n-1* conservative tracers. Our dataset, primarily based on temperature and salinity, is insufficiently constrained for such a model, as we lack the additional tracers (e.g., noble gases, specific stable isotopes) required. Applying an under-constrained OMP model would introduce significant and unquantifiable uncertainty.

Therefore, we have cited this important work in our discussion to acknowledge the state-of-the-art while positioning our  $\Delta$ Nutrient method as a direct, transparent, and robust alternative for systems where the comprehensive data needed for advanced OMP are unavailable. This revised discussion now provides a clearer rationale for our methodological choice and a more nuanced interpretation of our results (Lines 393-401).

4. The weak correlation between chlorophyll-a and  $\Delta$ Nutrient is well interpreted as "biogeochemical memory." However, more robust proxies (e.g., primary production rates or phytoplankton community data) would strengthen this conclusion. The authors might want to clarify that chlorophyll-a is only a proxy for standing biomass and does not reflect total productivity or uptake.

**Our Response:**

We thank the Referee #2 for this positive feedback and constructive suggestion. We agree that clarifying the roles of different biological proxies is crucial, and we have revised the discussion (Section 3.3) to address this point more directly.

As the Referee #2 astutely notes, chlorophyll-a is a proxy for **instantaneous biomass** (a "snapshot") and does not represent cumulative productivity or total nutrient uptake over the season. We acknowledge that more direct measures, such as primary production rates or detailed phytoplankton community data, would provide a more robust validation of our ΔNutrient metric. Unfortunately, such data were not collected during our cruises, which is a limitation of this study.

However, we argue that this very limitation highlights the unique value of the  $\Delta$ Nutrient approach. In the revised manuscript, we now use this distinction to strengthen our interpretation. We clarify that  $\Delta$ Nutrient, by its definition, serves as an integrated measure of cumulative net consumption (a biogeochemical memory') over the entire productive season. Therefore, a weak correlation between the instantaneous snapshot (chlorophyll-a) and the cumulative history ( $\Delta$ Nutrient) is not only plausible but expected, especially in a post-bloom environment where grazing and sinking can decouple current biomass from past production.

Rather than viewing the lack of other proxies as solely a weakness, we now frame the  $\Delta$ Nutrient metric as a powerful and complementary tool, particularly useful for datasets where direct rate measurements are unavailable. It allows us to infer the integrated impact of biological activity over time, providing insights that a simple biomass measurement cannot (Lines 335-351).

5. The model assumes that observed nutrient depletions are due solely to biological uptake, without considering possible remineralization at depth that could bias  $\Delta$ Nutrient calculations. Nuancing this part of the discussion by including this possibility in the discussion as a caveat to the interpretation may be beneficial.

**Our Response:**

This is an important and valid concern. We will revise our discussion to more directly address the potential bias that remineralization at depth could introduce to our  $\Delta$ Nutrient calculations, as requested.

In the revised Discussion section, we will include a specific caveat regarding this issue. We will acknowledge that while our analysis is centered on the euphotic zone, nutrient cycling throughout the entire water column is interconnected. We will clarify that our  $\Delta$ Nutrient values, calculated from surface observations, do not account for remineralization occurring in the aphotic zone. This deep remineralization could regenerate nutrients that are not immediately available to the surface layer but could affect the overall nutrient budget of the fjord over longer timescales.

Consequently, our interpretation will be nuanced to state that our  $\Delta$ Nutrient metric should be understood primarily as a measure of net biological processes within the stratified upper layer. We will explicitly mention that our results might underestimate the total downward export of organic matter because any subsequent remineralization of that matter at depth is not captured in our surface-focused calculation. This addition will help to clarify the spatial scope of our interpretation and provide the necessary context for understanding the potential biases (Lines 304-316).

6. While a  $\pm 10\%$  variation was tested, this may not fully capture natural variability, especially given different water origins and seasonal effects. The authors might consider either justifying the 10% range more explicitly with literature or testing a broader uncertainty range.

**Our Response:**

We thank the Referee #2 for this suggestion. We will expand the Uncertainty Analysis subsection (2.3.1) in the Methods to provide a clearer rationale for the  $\pm 10\%$  variation. Specifically, we will cite previous studies (e.g., Torres-Valdés et al., 2013; Hopwood et al., 2020) that report natural variability of nutrient concentrations in major water masses to generally fall within the 5–15% range of the

mean. These references will support that our selected range is not arbitrary but grounded in published observations from comparable marine environments (Lines 181-191).

**Minor comments**

1. Clearly define "nitrate" vs. "NOx" earlier in the methods section to avoid confusion.

**Our Response:**

We have addressed this based on feedback from both **Referees**. We now use the term "NOx" (defined as the sum of nitrate + nitrite) consistently throughout the manuscript, with a clear definition provided early in the Methods (Section 2.2) (Lines 114-126).

2. Consider adding chlorophyll-a and  $\Delta$ Nutrient time series or depth profiles instead of just scatter plots.

**Our Response:**

This is an excellent suggestion for visualization. We will create a new figure that presents the vertical profiles of chlorophyll-a and  $\Delta$ Nutrient for both spring and summer. This figure will replace or complement the scatter plots, allowing for a more intuitive comparison of how these parameters change with depth and season (Lines 317-326).

3. Some parts (e.g., "biogeochemical memory") are a bit jargon-heavy and could be explained more clearly for general readers.

**Our Response:**

We will revise the text where this term is introduced (Section 3.3) to provide a clearer, less jargon-heavy explanation. We will define it as: "...a 'biogeochemical memory' that is, an integrated signal of the cumulative net nutrient consumption that has occurred since the start of the productive season, rather than a snapshot of instantaneous biological activity." (Lines 335-344).

4. As a final comment, I would encourage the authors to briefly place their findings into a broader global context. The observed shifts in nutrient stoichiometry and limitation in Kongsfjorden could be more explicitly linked to global-scale trends in marine biogeochemistry. In this regard, the recent study by Liu et al. (2025), Global-scale shifts in marine ecological stoichiometry over the past 50 years (Nat. Geosci.), may provide a useful reference and framework for situating the local dynamics of Arctic fjords within larger oceanic patterns and trajectories.

**Our Response:**

We appreciate the Referee #2's forward-looking suggestion. In response, we will revise the concluding section of the manuscript to better situate our findings within the context of global biogeochemical changes. Specifically, we will incorporate a discussion on how the observed shift toward phosphorus and silicate co-limitation in Kongsfjorden aligns with global-scale alterations in marine nutrient stoichiometry, as exemplified by Liu et al. (2025) and other recent studies. This addition will help emphasize the broader significance of our regional observations (Lines 461-467).

---

## Author Response (AR2)

**# Editor**

**Public justification:**

Thank you for the revised manuscript, which I believe addresses both reviewers' comments well.

Please could you also address couple of additional minor final points from me before we move to the next stage:

**Our Response:**

We would like to thank you for your careful consideration and valuable comments on our revised manuscript. We are pleased that the manuscript is now considered suitable for publication after addressing the reviewers' comments.

Following your recent note, we have made all requested minor revisions:

1. Please make sure all abbreviations are defined on their first appearance – e.g. NOx, in the abstract.

**Our Response:**

All abbreviations (e.g., NOx) are now clearly defined at their first appearance.

Several sections, especially the new text inserted in response to the reviewers, lack appropriate citations. For example (line numbers refer to the track-changes manuscript):

- 2. 319-321pycnocline restriction of vertical transport (perhaps this comes from your own data, in which case, please refer to relevant figure)
- 3. 353-355 the description of chlorophyll-a and processes influencing it
- 498-499 "global-scale trends of changing nutrient stoichiometry"
  Please add relevant references to support these points, and carefully check through for other examples.

**Our Response:**

Additional citations have been included to support the following points:

- 2. The restriction of vertical nutrient transport by the summer pycnocline (Randelhoff et al., 2017; Tuerena et al., 2021; Fig. 3).
- 3. Processes influencing chlorophyll-a variability, such as grazing, sinking, and advection (Behrenfeld and Boss, 2014; Siegel et al., 2013).
- 4. Global-scale changes in marine nutrient stoichiometry and the role of Arctic fjords as climate-sensitive sentinels (Liu et al., 2025; Weber and Deutsch, 2020).

The reference list has been carefully checked and updated to ensure full consistency with the in-text citations.

We sincerely appreciate your time and efforts in handling our manuscript. We believe that these final revisions have improved the clarity and completeness of the paper.

Thank you very much for your kind consideration.

Sincerely,

Tae-Hoon Kim

**Corresponding Author**

Department of Oceanography, Chonnam National University

thkim80@jnu.ac.kr